# Evaluation and Treatment of Mild Traumatic Brain Injury: The Role of Neuropsychology

**DOI:** 10.3390/brainsci7080105

**Published:** 2017-08-17

**Authors:** Carolyn Prince, Maya E. Bruhns

**Affiliations:** 1JFK Johnson Rehabilitation Institute, Center for Brain Injuries, Edison, NJ 08820, USA; 2Alta Bates Summit Medical Center, Oakland, CA 94609, USA; BruhnsM@sutterhealth.org

**Keywords:** mTBI, concussion, PCS, neuropsychology, cognitive rehabilitation

## Abstract

Awareness of mild traumatic brain injury (mTBI) and persisting post-concussive syndrome (PCS) has increased substantially in the past few decades, with a corresponding increase in research on diagnosis, management, and treatment of patients with mTBI. The purpose of this article is to provide a narrative review of the current literature on behavioral assessment and management of patients presenting with mTBI/PCS, and to detail the potential role of neuropsychologists and rehabilitation psychologists in interdisciplinary care for this population during the acute, subacute, and chronic phases of recovery.

## 1. Introduction

In 2013, an estimated 2.5 million traumatic brain injury-related emergency department (ED) visits occurred in the United States [1]. Although estimates across analyses vary, it is generally thought that 75%–90% of these injuries would be classified as mild [2,3]. These percentages likely underestimate the total number of mild traumatic brain injuries (mTBI) since patients do not always present to the ED following a mTBI, with some patients following up with general practitioners and others not seeking any medical care [2,4]. As a result, a high percentage of mTBIs in the United States and worldwide may go underdiagnosed or unidentified. The purpose of this article is to detail the role of neuropsychologists and rehabilitation psychologists in the interdisciplinary care of patients with a history of mTBI. Our review focuses primarily on civilian adults who have sustained a mTBI, since the additional factors associated with assessment and treatment of children, adolescents, veterans, and/or athletes is beyond our intended scope. Despite this population focus, much of the information covered is likely generalizable across the mTBI population at large. In this review, we aim to provide education on neuropsychological evaluation and treatment of this often underdiagnosed and underserved population.

## 2. Defining Mild Traumatic Brain Injury

Adding to the complication of the likely underdiagnosis of mTBI is the lack of an interdisciplinary consensus regarding what constitutes a mTBI [4,5,6]. The American Congress of Rehabilitation Medicine (ACRM) was the first to establish diagnostic criteria of mTBI as involving “a traumatically induced physiological disruption of brain function, as manifested by at least one of the following: i) any period of loss of consciousness; ii) any loss of memory for events immediately before or after the accident; iii) any alteration in mental state at the time of the accident (e.g., feeling dazed, disoriented, or confused); and iv) focal neurological deficit(s) that may or may not be transient; but where the severity of the injury does not exceed the following: loss of consciousness of approximately 30 min or less; after 30 min an initial Glasgow Coma Scale (GCS) of 13–15; and posttraumatic amnesia (PTA) not greater than 24 h” [7] (p. 86). In their report to Congress, the US Centers for Disease Control and Prevention (CDC) posited a comparable, though less specific conceptual definition of mTBI as “any period of observed or self-reported: transient confusion, disorientation, or impaired consciousness; dysfunction of memory around the time of injury; loss of consciousness lasting less than 30 min” as well as “observed signs of neurological or neuropsychological dysfunction” [2] (p. 2). More recently, the Word Health Organization (WHO) task force on Mild Traumatic Brain Injury put forth a definition based on a review of the literature that varied from the ACRM diagnosis by simplifying the classification of altered mental status to “confusion or disorientation” and changing the “focal neurological deficit(s)” criteria of the ACRM definition to: “Other transient neurological abnormalities, such as focal signs, seizure, and intracranial lesion, which are not requiring surgery.” In addition, the WHO definition allows for the GCS score of 13–15 to be assessed after the typical 30-min timeframe, which accounts for a possible delay in assessment by a qualified healthcare provider [8] (p. 115). The lack of consensus in terminology complicates matters further, with the research literature using terms like concussion, mild head trauma, and mild head injury interchangeably. For clarity, this review will use the term mTBI exclusively.

## 3. Acute Identification and Evaluation of Mild Traumatic Brain Injury

In addition to the lack of a standard definition of mTBI, there is much variability in acute medical management of this common condition. In their evaluation of 41 guidelines related to mTBI, Peloso and colleagues [9] only categorized three as being evidence-based and reported that “in the absence of clear evidence, experts frequently disagree” [9] (p. 111). Blostein and Jones [10] surveyed 35 level I trauma centers in the United States regarding their evaluation and discharge of patients with suspected mTBI. They found that less than half of the centers had a standardized protocol in place for evaluating all patients with suspected mTBI. Foks and colleagues [11] found a similar lack of consistency in mTBI evaluation and management when they surveyed 71 neurotrauma centers in Europe and Israel. Powell and colleagues [12] found that over half of the 197 patients identified as having a mTBI by study personnel were not documented with that diagnosis by medical personnel in the ED. Within the Veteran population, Pogoda and colleagues [13] showed that clinical judgment differed from ACRM-based criteria for mTBI history in 24% of the cases seen for a comprehensive TBI evaluation, with the majority of these disagreements indicating that clinician judgment on mTBI diagnosis was inconsistent with ACRM-based criteria (Clinician N/ACRM Y). This outcome of Clinician N/ACRM Y reportedly occurred more often when veterans reported higher affective symptoms accompanied by lower reported cognitive and physical symptoms. The lack of consistent guidelines regarding acute ED evaluation and management of patients suspected as having sustained a mTBI likely contributes to the estimates that “50%–90% of patients with mTBI often go unidentified or undiagnosed in the hospital ED” [14] (p. 272). Patients who go undiagnosed may be at a higher risk for a “complicated recovery” [12] (p. 1554) because they are not provided with psychoeducation regarding possible consequences of mTBI and the expected recovery trajectory [12].

Even in the instance of a positive diagnosis of mTBI in the ED, many researchers have shown a lack of standardized guidelines regarding when to hospitalize, when to discharge home from the ED, and when to make a referral for outpatient follow-up [11]. Patients who are discharged directly from the ED are often expected to have a better recovery than patients who require hospitalization after mTBI; however, research reported by de Koning and colleagues [15,16] indicates that one in five patients who were directly discharged from one of three level-I trauma centers after mTBI had unfavorable outcomes at six months. Further, only a quarter of these patients followed up with an outpatient neurologist within the first six months of injury due to persisting symptoms. These unfavorable results may be due, in part, to the lack of clear guidelines regarding outpatient follow-up after direct discharge from the ED. Similarly, Foks and colleagues found that the majority of patients with a history of mTBI in Europe do not receive routine follow-up care [11]. Contributing to these concerns is the finding that many discharge instructions failed to address the possibility that patients may develop persisting post-concussive symptoms [4]. Overall, these findings suggest a need to develop more clear guidelines regarding discharge instructions and psychoeducation for all patients diagnosed with a mTBI, regardless of whether or not they required hospitalization. Further, additional research is needed to provide more clarification as to when and for whom follow-up care is appropriate.

## 4. Mild Traumatic Brain Injury Symptoms

As discussed above, there is a dearth of information provided to patients with mTBI discharged from the ED regarding symptom expectations. During the acute and subacute phases of recovery from a mTBI, patients generally report symptoms that fall into one of three symptom clusters: somatic (e.g., physical and/or sensory), cognitive, and affective (e.g., emotional). Commonly reported somatic symptoms include headache, sleep disruptions, dizziness, nausea, visual disturbance, photophobia, and phonophobia. Common cognitive symptoms include problems with attention and memory, slow processing speed, difficulty multitasking, increased distractibility, losing one’s train of thought, and feeling foggy. Affective symptoms often reported by patients with mTBI include increased irritability, emotional lability, anxiety, and depression [17,18]. Fatigue is a frequent complaint after mTBI. Research regarding fatigue suggests that it is a multidimensional symptom, with many factors contributing to and exacerbating fatigue, including somatic symptoms, sleep disturbance, cognitive exertion, chronic situational stress, and mental health [19,20,21,22].

Similar to the cycle of symptom exacerbation observed in fatigue, many somatic, cognitive, and affective symptoms following mTBI interact with and exacerbate each other. Kay and colleagues [23] described a dysfunctional feedback loop that may result in the maintenance of numerous symptoms even after medical signs resolve. For example, when cognitive difficulties during the early phase of recovery co-occur with pain symptoms or emotional factors, a feedback loop may develop whereby the pain and emotional factors start to elicit secondary cognitive complaints. Over time, the interaction between these symptoms strengthens to the point that persisting pain and emotional factors will continue to elicit subjective cognitive complaints even after the primary cognitive complaints have resolved. Other researchers have identified the associations between mTBI symptoms as well. Wood and colleagues [24] found that patients with mTBI history who scored higher on measures of anxiety sensitivity and/or alexithymia tended to score higher on the Rivermead post-concussion symptom questionnaire at two weeks post injury than patients scoring lower on these measures. In a different study, cogniphobia, defined as “fearful avoidance of a specific headache trigger, mental exertion” [25] (p. 1), was found to be associated with worse memory test performance in a group of patients with mTBI history who reported more severe post-traumatic headaches [25]. The development of these dysfunctional feedback loops “may serve to maintain a variety of symptoms beyond the resolution of the original organic deficit” [26] (p. 552).

## 5. The Controversy regarding Persisting Post-Concussive Symptoms

While the majority of patients with mTBI history are asymptomatic within a couple of weeks post-injury, a small minority (10%–20%) of patients continue to report detrimental symptoms for months and even years post-injury [5]. This group of patients with unfavorable outcomes following mTBI is sometimes termed the “miserable minority” [26] (p. 551), and the condition is termed post-concussive syndrome (PCS). The outcomes of this patient group have generated much controversy, with the emergence of two polar opinions among professionals as to whether these persisting symptoms are the result of neurogenic factors (e.g., neurological residuals of the original mTBI) or psychogenic factors (e.g., pre-morbid psychopathology or personality characteristics) [5,27,28]. Others take the more balanced perspective that these two opinions are “complimentary and capable of being integrated” [27] (p. 1120).

The controversy regarding this patient group has resulted in many researchers attempting to determine the etiology of their symptoms. Numerous studies have reported that patients with a premorbid psychopathology history are more likely to report post-concussion symptoms beyond one month post-injury [17,29,30]; however, the same researchers have shown that premorbid psychopathology no longer predicts post-concussion symptoms at the one year mark [29] and others have shown no association between premorbid psychopathology and post-concussion symptom reporting [31]. Premorbid personality traits, including perfectionism, grandiosity, individuals with unmet dependency needs, and borderline traits, have been identified as possible risk factors for development of PCS after mTBI [23,26]. Patients who present to the ED with moderate to severe somatic complaints (e.g., headache, light sensitivity, blurred vision) tend to have less favorable outcomes at 12 months post-injury [32]. Similarly, patients who are highly symptomatic at one month often continue to be highly symptomatic at one year [29]. Injury characteristics like mTBI severity and extra-cranial bodily injuries predict future PCS symptom development [14,29]. Biological factors that have been identified in prolonging mTBI recovery include genetics, older age, female sex, prior mTBI history, and poorer physical health, to name a few [14,29,30,32,33]. Various psychosocial factors thought to contribute to poorer outcome post-injury include: disability or unemployment at time of injury, marital status, education level, occupational skill level, cognitive reserve, financial and recreation setbacks secondary to injury, involvement in litigation, substance abuse, and other life stressors [29,30,32,33].

Uomoto and Fann [34] found that symptomatic patients with mTBI history tended to overestimate their injury severity level and their residual cognitive impairment when compared with patients who had sustained a moderate or severe TBI. In addition, the patients with mTBI history tended to “view their injury as having a more global impact across important areas of their life” [34] (p. 336). Iverson and colleagues [35] evaluated the so-called “good old days” bias (p. 17) with regard to patients with a history of mTBI and found that, in comparison to healthy adults, these patients report fewer, less severe premorbid symptoms. Furthermore, they found that patients who failed performance validity testing (TOMM) reported fewer premorbid symptoms and more debilitating post-concussion symptoms than patients with mTBI history who passed validity testing [35].

The literature presents different viewpoints on neurogenic versus psychogenic causes of PCS, and to date the reasons for symptom persistence after mTBI are not fully understood. However, many psychologists engage in patient-centered care, and therefore are able to evaluate, target, and treat the individual needs of each patient experiencing PCS, regardless of symptom etiology. In the sections that follow, the authors will provide a brief review of the evaluation and intervention methods available for patients with mTBI history within the field of neuropsychology, rehabilitation psychology, and cognitive rehabilitation.

## 6. Neuropsychological Evaluation Following Mild Traumatic Brain Injury

The purpose of a neuropsychological evaluation is to assess the cognitive and functional deficits resulting from a neurological disorder or injury. As part of a comprehensive evaluation, the neuropsychologist conducts a thorough clinical interview that reviews the presenting condition and associated symptoms as well as premorbid patient characteristics and psychosocial history factors that may be contributing to the clinical presentation. Information from individuals who know the patient well is often pursued to gain additional insight regarding the patients’ pre- and post-morbid behaviors. The information gathered from these clinical interviews is integrated with the results of cognitive testing and self-report measures to construct a holistic view of the etiology of the patient’s present complaints. Perhaps the most important phase of the neuropsychological evaluation is when the patient, the patient’s family, and the referring provider are given feedback and psychoeducation regarding the evaluation findings and provided with recommendations for treatment.

This detailed, holistic approach to evaluating the patient is especially helpful when it comes to working with patients recovering from a mTBI since, as discussed above in Section 5, there are many confounding factors that are likely contributing to and exacerbating the symptomatic presentation. If a neuropsychological evaluation is conducted shortly after a mTBI, the treatment team can be informed of any factors that may prolong the patient’s recovery and develop interventions to help mitigate these factors. In addition, earlier evaluation ensures that the patient receives psychoeducation regarding mTBI and the expected recovery trajectory, the importance of which is discussed in more detail below in Section 7.1.

### 6.1. Cognitive Dysfunction after a Mild Traumatic Brain Injury

After a mTBI, cognitive dysfunction is often seen in the domains of attention, processing speed, executive functions, and/or memory, although there is differential recovery across these domains over time [23,27]. The observed deficits can be relatively subtle and are influenced by numerous factors, including injury severity, time since injury, and the specific neuropsychological measures used [36]. With regard to the influence of time since injury, Karr and colleagues conducted a systematic review of meta-analyses and found that multiple studies report a return to cognitive baseline by 90 days post-injury [37]; however, some patients who have developed a more chronic PCS continue to show neuropsychological impairments on testing [25,33]. There is also great variability across neuropsychological tests with regard to their ability to capture cognitive deficits secondary to mTBI [37].

### 6.2. Neuropsychological Evaluation of Mild Traumatic Brain Injury

As discussed above a comprehensive neuropsychological evaluation consists of a clinical interview of the patient and collaterals, when available; a thorough test battery that capture all cognitive domains; and self-report measures to evaluate the patient’s mood and current symptoms. With regard to an evaluation of a patient with a mTBI, thorough assessment of attention, processing speed, executive functions, and memory is necessary to capture any current cognitive deficits. Mood measures should assess for symptoms of depression, anxiety, irritability, and emotional lability. Finally, inclusion of a self-report measure that captures symptoms that are common after a mTBI is recommended since the patient may have trouble relaying all of their symptoms during the clinical interview. These measures also provide the patient with a method for ranking the severity of the symptoms at the time of the evaluation. When interpreting cognitive test performance, the neuropsychologist looks for patterns of deficits across the tests, since one test score indicative of impairment does not necessarily translate into diagnosis of a cognitive deficit. Consideration is also given to the “sterile,” distraction-free testing environment, which may mitigate the “real-world” impact of cognitive dysfunction following mTBI that the patient experiences in his or her daily life [38]. Table 1 details a sample mTBI battery used by one of the authors (C.P.) in her evaluations of patients with mTBI history; however, this battery is only given as an example of a comprehensive neuropsychological evaluation. Providers are encouraged to develop batteries based upon the individual needs of their patients as well as the referral question.

## 7. Neuropsychological Intervention Following Mild Traumatic Brain Injury

While several decades of research have established an increasingly clear picture of typical mTBI symptoms and the complexity of persisting PCS, research on effective interventions for this population has yielded mixed results, and therefore has not yet supported professional consensus on mTBI treatment. Despite the relative dearth of well-controlled prospective research studies in this population, the existing literature does provide preliminary support for a number of behavioral interventions for management and amelioration of symptoms following mTBI. Given the complexity and variability of presenting problems after mTBI as detailed above, treatment must account for multiple factors including cognitive, emotional, and somatic symptoms. Neuropsychologists and rehabilitation psychologists are uniquely suited to manage care for these concerns holistically, given our expertise on psychological and cognitive functioning, as well as an established presence in Neurology and Rehabilitation departments. The following section provides an overview of neuropsychological interventions for individuals who have sustained a mTBI.

### 7.1. Early Intervention in the Acute Phase Following Mild Traumatic Brain Injury

Most individuals who sustain a mTBI spontaneously recover fully within the first few weeks or months, but a significant minority continue to experience persistent symptoms for months or years following injury [39,40]. A small but significant body of research has addressed the question of whether psychoeducational and supportive interventions in the acute phase after mTBI can prevent progression to persistent PCS. These interventions are based on the theory that persistent PCS is associated with attribution of symptoms to the mTBI and negative expectations about recovery [41]. Such interventions typically involve education on post-concussive symptoms, reassurance and education on the expectation for complete recovery, and guidance regarding rest and gradual resumption of typical activities. Psychoeducational early interventions have the strongest empirical support of any post-mTBI interventions, with several systematic reviews concluding they are well supported [42,43,44]. One recent systematic review with stringent methodological exclusion criteria [45] found that only two studies in the entire corpus of research on mTBI treatment met its standards; one supporting telephone-based early educational intervention [46], and one supporting recommendations for bed rest in the acute recovery phase [47].

Acute care for mTBI is typically provided by medical professionals, including ED staff and primary care physicians [48,49]. Individuals are often not referred for consultation with a neuropsychologist until their symptoms persist beyond the typical period of spontaneous recovery. However, it is our opinion that the literature on early intervention after mTBI supports earlier consultation with psychology or neuropsychology, particularly for individuals at highest risk for developing persistent PCS. Psychological distress and psychosocial stressors are associated with greater likelihood of progression to persistent PCS [23,24,25,26], and neuropsychologists and rehabilitation psychologists are well equipped to provide early educational interventions that address the interplay of cognitive, emotional, and somatic symptoms. Psychologists also can provide psychotherapeutic intervention when appropriate, including validation of the individual’s experience and instillation of hope. When a neuropsychological evaluation is feasible, neuropsychologists can tailor preventative education to the individual’s neuropsychological profile. In short, early consultation with Neuropsychology may be beneficial to more comprehensively address the needs of the subset of patients who are most at risk for persisting PCS. Current research supports early psychoeducational and supportive services after mTBI [42,43,44]; we would argue for broad application of such services, as these interventions are brief, relatively low cost, and may prevent progression to PCS, hence decreasing the individual burden of disability as well as the societal burden of more resource-intensive services.

### 7.2. Neuropsychological Treatments for Persisting Post-Concussive Symptoms

A relatively smaller amount of research has examined interventions for the significant minority of individuals who continue to experience post-concussive symptoms for months or years. Awareness of PCS has increased in recent decades, due in part to high frequency of mTBI in the conflicts in Iraq and Afghanistan, as well as well-publicized cases involving athletes. The research literature on specific treatments for PCS is limited, and systematic reviews have consistently concluded that the existing literature is limited by methodological inconsistency [43,44,45]; however, clinicians can also refer to the much larger literature on treatment of specific symptoms common to PCS, such as cognitive rehabilitation for attention deficits, or cognitive behavioral therapy for symptoms of depression and anxiety. Thorough discussion of specific interventions is beyond the scope of this review, but we strive to provide a brief overview of the types of treatments for PCS that may be provided by neuropsychologists and rehabilitation psychologists.

Cognitive Rehabilitation: Cognitive rehabilitation comprises an eclectic set of therapeutic approaches that are tailored to the individual’s neuropsychological profile and functional goals [50]. The first step in treatment planning is a thorough neuropsychological assessment. Based on this assessment, the clinician may draw from a number of therapeutic approaches, incorporating not only strictly cognitive interventions but also emotional, behavioral and social interventions as needed. Broadly, cognitive rehabilitation interventions can be categorized as “bottom-up” interventions, which build or restore basic skills using rote practice, or “top-down” interventions, which use metacognitive skills, or “thinking about thinking,” to promote effective self-management of cognitive difficulties. Top-down approaches can be further subdivided into internal strategies, such as self-monitoring and self-regulation, and external strategies, such as reminders and organizational systems. In clinical practice, treatment typically involves a combination of these approaches. The following paragraphs provide a brief overview of cognitive rehabilitation interventions relevant to mTBI and PCS. Clinicians are advised to refer to the ACRM Cognitive Rehabilitation Manual [51] for more comprehensive guidelines on the interventions outlined here.

Individuals with persistent PCS often present with cognitive deficits in attention regulation, executive functions, and memory [20]. Systematic reviews of cognitive rehabilitation in PCS have reached varying conclusions, and no true meta-analyses have been possible to date due to substantial variability in interventions, outcome measures, and study designs. The most recent systematic review at the of time of this writing reviewed interventions for mTBI in military/veteran populations and found good support for the efficacy of cognitive rehabilitation [43]; similarly, a 2009 consensus conference on services for military service personnel and veterans with mTBI history strongly endorsed cognitive rehabilitation interventions [52]. The civilian literature on mTBI remains limited, and most systematic reviews have not resulted in firm support for cognitive rehabilitation interventions in treatment of cognitive symptoms in mTBI/PCS [44,53]. However, there exists a substantial literature on cognitive rehabilitation interventions for attention, executive functions, and memory, which can be assumed to be applicable to individuals with PCS, according to their neuropsychological profile.

Deficits in attention regulation (e.g., working memory, multi-tasking, distractibility) are among the most common features of PCS. Based on systematic reviews of the cognitive rehabilitation literature, Cicerone and colleagues [54] recommend cognitive remediation of attention deficits after TBI as a “practice standard”. Specifically, “remediation of attention deficits after TBI should include direct attention training and metacognitive training to promote development of compensatory strategies and foster generalization to real world tasks” [54] (p. 521). Computer-based interventions are classified as a “practice option” in conjunction with therapist-guided treatment to promote functional application of skills. Evidence-based treatments for remediation of attention include Attention Process Training [55], n-back working memory remediation [56], and Time Pressure Management [57]. These interventions should be implemented with explicit focus on the development of proactive compensatory strategies to manage attentional resources.

Remediation of executive functions may have especially far-reaching effects, as executive skills (e.g., planning, prioritizing, problem-solving) are applicable across rehabilitation disciplines and to all aspects of daily living. The Cicerone et al. reviews [54,58] recommend metacognitive strategy training, including self-monitoring and self-regulation skills, as a practice standard; this approach may be applied to self-regulation of cognition, emotion, and behavior, and is useful as a component of other rehabilitation interventions. Training in problem-solving strategies is a “practice guideline” for executive dysfunction after TBI, and group-based interventions are a “practice option”. Most structured treatments for executive functions follow a four-step sequence: building awareness; anticipation of difficulties and planning accordingly; task execution and self-monitoring; and self-evaluation following the task [51]. Structured interventions include goal management training [59], problem-solving training [60], and the Cognitive Orientation to Occupational Performance (Co-Op) [61]. The specific elements within each of these steps vary, but the process of anticipation, execution, and evaluation is common across interventions.

Difficulties with memory are among the most common subjectively reported cognitive problems. In individuals with mTBI, memory difficulties often represent a “downstream” effect of attentional and/or executive deficits impacting the acquisition and retrieval of memories, rather than a true memory encoding or retention deficit. Therefore, memory may be positively impacted by attentional and executive training. For further specific rehabilitation of memory, the Cicerone et al. reviews [54,57] recommend memory strategy training as a practice standard for mild memory impairments after TBI, including both internal and external strategies. Internal, or metacognitive, memory strategies include association techniques such as visualization or “peg” methods associating new material with well-known information, and organizational techniques such as acronym/rhyming mnemonics or contextualization/chunking of information [62]. External strategies include memory notebooks, strategically placed external cues/reminders, as well as ever-evolving use of smartphones and other technological aids.

It is important to note that attention, executive functions, and memory are mutually interdependent processes [50,58], and many interventions explicitly target several cognitive domains simultaneously. Combined attentional-executive interventions typically consist of metacognitive strategy training grounded in the patient’s functional goals, and may also incorporate direct skill training, though isolated use of rote skill training without metacognitive skills is not recommended [58]. Integrated treatment of executive skills and attention regulation has been shown to improve functional outcomes [63,64] as well as modulate prefrontal cortical activity as observed through fMRI [65].

Psychotherapy: Individuals with PCS typically present with emotional dysregulation, and often meet diagnostic criteria for clinical psychological disorders including depression, anxiety, post-traumatic stress, and substance use disorders [40]. Neuropsychological assessment should include screening measures for mental health conditions at a minimum, and may include more comprehensive psychological assessment if necessary. In some cases, psychological treatment may be necessary either as a prerequisite to, or concurrent with, other interventions, since individuals in acute psychological distress may have difficulty participating actively in other rehabilitation interventions, and emotional distress may exacerbate other cognitive and somatic symptoms.

One systematic review on behavioral health interventions post-mTBI [66] examined four categories of psychological interventions: cognitive-behavioral therapy (CBT) for PCS; educational interventions; cognitive rehabilitation interventions with a psychotherapeutic element; and mindfulness/relaxation training programs. They found adequate support for CBT, and concluded that the literature on integrated rehabilitation/psychotherapeutic programs and mindfulness/relaxation was too limited at that time to make recommendations. In a recent review of post-concussive treatment among veterans and military service personnel, several studies were identified that showed promising early results for the feasibility of psychotherapeutic interventions in individuals with mTBI, though recommendations were limited by small sample sizes and lack of control groups [51]. Some individuals may experience decrements in perceived quality of life and life satisfaction after mTBI [67], indicating a need for psychotherapeutic support for coping and adjustment.

Integrated Treatment Options: An increasing number of brain injury treatment programs are structured to integrate psychoeducation, cognitive rehabilitation, and psychotherapy, “operating from an assumption of comorbidity” [43]. From a neuropsychological perspective, this approach makes logical sense, as changes in the brain have broad impacts across cognitive, emotional, social and behavioral functioning. Particularly, deficits in executive functions have broad impacts upon the awareness, monitoring, and regulation of thoughts, feelings, and behaviors. Accordingly, integrative treatment programs apply the core concepts of metacognition and self-regulation to all aspects of rehabilitation. The most recent systematic review of cognitive rehabilitation by Cicerone and colleagues identifies “comprehensive-holistic neuropsychological rehabilitation” as a distinct category of post-TBI intervention, and it is recommended as a practice standard for individuals with moderate to severe TBI [54]. Research on comprehensive-holistic interventions with mTBI is limited, but a few studies have shown promising outcomes [63,68,69].

Beyond comprehensive neuropsychological intervention, care for individuals with persistent PCS often involves interdisciplinary collaboration across specialties. In a rehabilitation setting, this may include: physical therapists, who may address vestibular/balance dysfunction and neck/back problems; occupational therapists, who treat vision problems, upper extremity coordination, and provide compensatory strategies for activities of daily living; and speech therapists, who often address attentional and executive aspects of communication such as complex comprehension and verbal organization. Depending on the patient’s functional needs, the treatment team may also include vocational, educational, or recreational specialists. Some patients may also benefit from additional alternative medicine or complementary therapies such as acupuncture [70] or meditation [71]. The treatment team should be assembled to address the patient’s individualized needs while integrating care for cognitive, emotional, and somatic concerns.

### 7.3. Tracking Progress and Outcomes

Measuring progress and outcomes from neuropsychological rehabilitation for mTBI, whether for clinical or research purposes, is challenging given the variability of baseline symptoms, the subjectivity of many common presenting problems, and the lack of a reliable relationship between objective measures (e.g., neuropsychological tests, neuroimaging), and subjective sense of progress or success. The fundamental goal of any rehabilitation intervention is to improve independence and quality of life, but there is not yet consensus in the field on how best to measure these subjective variables.

Regarding objective and standardized measures, formal neuropsychological re-evaluation can be especially helpful in the earlier stages of recovery, or after provision of direct cognitive remediation interventions, in order to track recovery and/or treatment efficacy. This can involve a full formal re-evaluation, or selective periodic re-administration of tests that are sensitive to change over time. Self-report instruments can provide a structured and normed measure of subjective symptoms, and repeat administration of self-report measures (or specific relevant items) can help to track the impact of the treatment on subjective concerns. Some research studies use generalized functional variables such as return-to-work as objective measures of real-world outcomes.

In clinical practice, the most meaningful gains are not necessarily captured well by neuropsychological tests or by broad variables such as return-to-work. Therefore, it is often both practical and therapeutic to identify client-centered treatment goals, which then serve both as a mechanism of change and as a way to measure progress. The process of setting realistic goals and benchmarks for progress can in itself be therapeutic, as it sets realistic expectations and promotes a sense of self-efficacy. One approach to setting clear, achievable personalized goals is Goal Attainment Scaling (GAS) [72], which can be incorporated into executive interventions such as goal management training [73]. Goal setting is a collaborative process, and must involve both client selection of relevant goals and therapist assistance in clarifying and defining realistic and measurable goals [74]. Attainment of goals defined at the onset of treatment, or lack of progress towards those goals, can provide a collaborative basis upon which to terminate or extend treatment. Clear documentation of goals and outcomes also serves to monitor treatment efficacy and contributes to the growing evidence base behind neuropsychological rehabilitation.

## 8. Conclusions

Proper identification and treatment of mTBI and post-concussive syndrome is challenging due to lack of consensus in the health care system regarding mTBI diagnosis and management, complexity of symptom presentations and comorbidities, heterogeneity with regard to symptom development and spontaneous recovery, and the dearth of prospective, controlled studies examining services for PCS. However, the existing literature does provide growing support for the efficacy of neuropsychological evaluations and interventions across stages of recovery from mTBI, from early preventative interventions to compensatory strategy training in the chronic phase. In order to provide comprehensive and patient-centered care, neuropsychologists and rehabilitation psychologists can be a valuable addition to the interdisciplinary teams caring for patients with a history of mTBI.

## Figures and Tables

**Table 1 brainsci-07-00105-t001:** Sample mTBI Neuropsychological Evaluation Test Battery.

Functional Domain	Neuropsychological Tests
Pre-Morbid Estimate	Test of Premorbid Functioning (TOPF)
Performance Validity	Reliable Digit Span
	California Verbal Learning Test-Second Ed. (CVLT-II): Forced Choice
Motor	Grooved Pegboard
Attention/Working Memory	Wechsler Adult Intelligence Scale- Fourth Ed. (WAIS-IV): Digit Span, Arithmetic
	Neuropsychological Assessment Battery (NAB): Orientation, Numbers & Letters
	Continuous Performance Test of Attention
	Paced Auditory Serial Addition Test (PASAT)
Processing Speed	Trail Making Test A
	WAIS-IV: Symbol Search, Coding
	Delis-Kaplan Executive Function System (D-KEFS) Color-Word Interference: Color
	Naming, Word Reading
Language	WAIS-IV: Similarities, Vocabulary, Information
	Animal Naming
	Boston Naming Test
	Complex Ideation
Visuospatial	WAIS-IV: Block Design, Matrix Reasoning
	NAB: Visual Discrimination
	Rey Complex Figure Test (RCFT): Copy
Memory	CVLT-II
	NAB: Story Learning, Shape Learning
	RCFT
Executive Functions	Trail Making Test B
	Controlled Oral Word Association Test (COWA-FAS)
	DKEFS Verbal Fluency: Category Switching
	DKEFS Color-Word Interference: Inhibition, Inhibition/Switching
	Tower of London-Second Ed.
	Wisconsin Card Sorting Test-64 Card Version
Mood Self-Report	Beck Depression Inventory-Second Ed. (BDI-II)
	Beck Anxiety Inventory (BAI)
	Patient Health Questionnaire: Somatic Symptom Scale
Neurobehavioral Symptom Self-Report	Neurobehavioral Symptom Inventory

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
