# Peer review of "Evaluation and Treatment of Mild Traumatic Brain Injury: The Role of Neuropsychology"

_brainsci, 2017, doi:10.3390/brainsci7080105_

Round 1

Reviewer 1 Report

This manuscript is a narrative review with the objective to highlight the role that neuropsychologists and rehabilitation psychologists can play in the evaluation and treatment of mild traumatic brain injury (mTBI).

Suggestions/revisions are listed below:

Major Compulsory Revisions

1.     Section 3, page 3, lines 98-100: The authors have overstated the literature findings here. I agree that findings suggest a need to develop more clear guidelines regarding discharge instructions, education in general, but I do not believe there is evidence to suggest we need follow-up care for all patients diagnosed with a mTBI. Most mTBI patients recover quickly without any follow-up care required. We need more high-quality research to identify the factors that are associated with the need for follow-up care. Recommending follow-up care for all post mTBI would put an unnecessary burden on the healthcare system.

2.     Section 7.1, page 7, lines 270-282: The authors state “It is well established that psychological distress and psychosocial stressors are associated with greater likelihood of progression to persistent PCS”; yet, only one weak study using a case series design is cited. There is no strong evidence corroborating any of the authors’ statements in this paragraph.

3.     Overall design: A narrative review does not involve any examination of the quality of the studies cited. Additionally, no comprehensive literature search is provided. Evidence of varying quality is used to justify author opinion. Scientifically, this is a weak approach to the summarization of evidence. Providing a clear literature search strategy and a methodological evaluation of the evidence cited would improve the scientific quality of the document drastically.

4.     Conflicts of Interest: A conflict of interest seems inherent given the manuscript argues for the involvement of a neuropsychologist to be standard of care for individuals with mTBI and at least one author is a neuropsychologist.

Author Response

Manuscript ID: brainsci-210894

Title: Evaluation and Treatment of Mild Traumatic Brain Injury: The Role of Neuropsychology

Brain Sciences Special Issue: Mild Traumatic Brain Injury (mTBI): Medical Treatments and Complications

Authors’ Note:

We thank the reviewers and assistant editor, Annie Zhang, for your time, careful review of our manuscript, and thoughtful and useful feedback, comments, and suggestions.

We have attempted to incorporate your feedback into the current manuscript when possible and have utilized track changes in Microsoft Word to highlight any changes that we have made to the original manuscript. We have not been able to address some of the concerns voiced by both reviewers regarding our overall design. The authors were invited by the guest editors of this Special Issue of Brain Sciences to provide an overview of the role that neuropsychologists can play in the treatment of individuals presenting with a mTBI history. Our intent with this article was not to provide a systematic review or meta-analysis of the current literature. Instead our intent was to provide education regarding the various roles that neuropsychologists can play in the treatment of individuals with mTBI history. With this intent in mind, and in response to some of our feedback, we have made revisions to our original submission that we hope will better convey our intended purpose.  We have also included individualized responses to all of the reviewers’ comments below. Thank you again for your time and consideration of our manuscript.

Responses to Comments:

Reviewer 1 Comments and Responses:

Comment 1: Section 3, page 3, lines 98-100: The authors have overstated the literature findings here. I agree that findings suggest a need to develop more clear guidelines regarding discharge instructions, education in general, but I do not believe there is evidence to suggest we need follow-up care for all patients diagnosed with a mTBI. Most mTBI patients recover quickly without any follow-up care required. We need more high-quality research to identify the factors that are associated with the need for follow-up care. Recommending follow-up care for all post mTBI would put an unnecessary burden on the healthcare system.

Response: Thank you. You make an excellent point regarding the appropriateness of follow-up care and the associated burden if we were to refer all patients with mTBI for follow-up. We have revised the final sentences of the paragraph to reflect the need for more clear guidelines regarding discharge instructions and psychoeducation. We have omitted the statement about follow-up care for all patients and have, instead, included the following sentence: “Further, additional research is needed to provide more clarification as to when and for whom follow-up care is appropriate.”

Comment 2: Section 7.1, page 7, lines 270-282: The authors state “It is well established that psychological distress and psychosocial stressors are associated with greater likelihood of progression to persistent PCS”; yet, only one weak study using a case series design is cited. There is no strong evidence corroborating any of the authors’ statements in this paragraph.

Response:  Thank you for this feedback. The association between psychological distress and PCS was reviewed in greater depth in section 5, and appropriate references have now been added to this section.  Regarding the remainder of the paragraph, you are right to point out that this paragraph shifts from a direct review of the literature (as in the previous paragraph), towards extrapolating from these finding to advocate for potential roles of neuropsychology.  This was not made clear in the first draft and changes have been made across the document to clarify which statements are direct supported by research and which statements include an element of author opinion. 

Comment 3: Overall design: A narrative review does not involve any examination of the quality of the studies cited. Additionally, no comprehensive literature search is provided. Evidence of varying quality is used to justify author opinion. Scientifically, this is a weak approach to the summarization of evidence. Providing a clear literature search strategy and a methodological evaluation of the evidence cited would improve the scientific quality of the document drastically.

Response: We agree in principle that a structured, systematic review would be a more rigorous scientific document than a narrative review.  However, the current paper was not intended to be and is not a thorough systematic review.  We were asked by the guest editors to provide a review of the various roles neuropsychologists may play in the evaluation and treatment of mTBI, in order to inform and educate consumers of this special issue.  This is a broad topic, and a complete structured review would be far beyond our scope for this paper.  We agree that the paper includes elements of author opinion, which we believe to be appropriate to the request for an educational/informational review.  Your point regarding varying quality of research is well taken, and we have made changes throughout the document to address quality of studies cited, as well as to clarify which statements are directly supported by empirical research and when we extrapolate from existing findings.  We also make many references to existing systematic reviews when they are available, and have highlighted topics on which there is a particular need for further research to support clinical recommendations.

Comment 4: Conflicts of Interest: A conflict of interest seems inherent given the manuscript argues for the involvement of a neuropsychologist to be standard of care for individuals with mTBI and at least one author is a neuropsychologist.

Response: We understand your concerns regarding the potential for a conflict of interest given that both authors are practicing in the field of neuropsychology, as a post-doctoral fellow and licensed professional respectively with regard to authorship. When looking at the definition of a conflict of interest provided by Merriam-Webster; however, we do not feel that our authorship of this manuscript fits the definition. A conflict of interest as defined by Merriam-Webster is: “a conflict between the private interests and the official responsibilities of a person in a position of trust.” Our goal with this manuscript was to provide an overview of the role that neuropsychologists can play in the management of individuals with mTBI and was intended to provide additional education to individuals who may be less familiar with our field of practice. We do not feel that this manuscript will result in any direct personal gain (e.g., our “private interests”) to the authors; however, we do feel that it is our role as neuropsychologists to offer additional education regarding our approach to evaluation and treatment of these patients. The same would be true of a surgeon detailing a specific surgical technique in a medical journal. In reviewing the manuscript in light of the feedback from both reviewers, we came across several instances of strong language advocating for the need for neuropsychological involvement in treating individuals with mTBI. In the current revision, we have attempted to tone down our message to one of providing education about the services that we can provide.

Reviewer 2 Report

Major conceptual comments.  Thank you for the opportunity to review this paper.  This manuscript discusses the importance of neuropsychological testing following mTBI, research, and clinical intervention approaches.  The main concerns are that there should be more discussion about the literature – references should be added where noted, and there needs to be more of an in-depth analysis of what the limitations of the literature are.  The tone of the article seemed quite negative towards research, and may distract from the intended message.  One can critically evaluate the literature by describing the findings and noting its strengths and weaknesses without editorializing.

Other Comments that must also be addressed and/or considered.

Abstract, Lines 14-15:  "is briefly reviewed" should be "are" briefly reviewed since the authors are discussing "Complications"

Abstract, Line 17: Delete "unfortunate" - it's implied.

Line 23:  "Estimates across analyses vary, however, it"  - suggested rewording: "Although estimates across analyses vary, it is generally..."

Line 24:  Remove "With that said, these estimates" and instead replace with "These percentages likely underestimate..."

Line 28:  What's the difference between an mTBI being "unidentified" vs. being "underdiagnosed?"  It would seem like a mild TBI being called a "bump on the head" would be an "underdiagnosis", (but could be unidentified), but not going to the doctor at all is "unidentified" because there's nobody being asked to make an identification.   What I'm inferring from your previous sentence about going to a general practitioner is that they don't have the expertise to diagnose a TBI, and therefore it's "underdiagnosed."  To make your sentences parallel, I would rearrange and say "underdiagnosed or unidentified" or just choose one of those terms depending on your intent.

Line 47, Capitalize so that it says "Congress."

Line 74:  Emergency Department can be abbreviated to ED since it's previously defined.

Line 77: I'd be careful about using the word "failed" because it seems unnecessarily harsh. Instead, I'd say something like "....clinician judgment on mTBI diagnosis was inconsistent with ACRM-based criteria...." 

Line 91 "Discharged" instead of "Discharge"

Line 143:  Similar to my previous comment, I would be more neutral in how something is presented.  Instead of saying "suffering" I would say something like "symptoms" or "health condition(s)"

Line 147:  Similar to previous comments, I'd choose another word to describe this situation.  Instead of saying "failed," I'd say something along the lines of "data not supporting an association" - otherwise it looks like there was something deficient with the researchers.

Lines 161-169, instead of saying "mTBI patients," I'd say "patients with mTBI history".  That phrase is used several times throughout this paragraph.

Lines 170-171 - "Despite the efforts of many researchers attempting to win the controversial debate..." Suggested rewording: "The literature offers different viewpoints on neurogenic versus psychogenic causes of PCS, and to date the reasons for symptom persistence is not fully known."  To say "etiology of symptom persistence" suggests a single source.

Lines 172-173: I'm not sure if this sentence is needed...it seems like a truism that anyone (even researchers!) would agree that appropriate care is needed for these or any other patients.

Lines 173-174: The paragraph starts off by talking about bickering researchers who probably agree that patients with PCS should get appropriate care, and then transitions into psychologists saving the day.  I'd suggest that your audience may be turned off by this approach, especially since may psychologists are also researchers.  Eliminate the word "fortunately", and I suggest saying something like "As many psychologists engage in patient-centered care, regardless of symptom etiology, patient-specific factors can be evaluated...."

Lines 170-176 - I would strongly urge the authors to reconsider their approach to this paragraph.  The beginning of the paragraph makes it seem as if researchers are fighting among themselves trying to prove superiority over an age-old issue that will likely not get resolved anytime soon, and the authors then concede that researchers probably think that people with health conditions should have be treated with the right kind of care.  But, as researchers bicker amongst themselves, fortunately there are psychologists who are above the fray and get to what's important - treating symptoms and taking into account multiple factors so they can make informed decisions about treatment.  The tone of the paragraph makes it seem the authors are pitting researchers against clinicians, and these populations, which at many times complement each other, often also intersect. 

Line 178:  Add "history" after mTBI

Line 224: "Since one impaired test score" sounds a little funny; what about "Since one test score indicative of impairment does not necessarily translate...."

Line 226:  Eliminate "to the fact" and state something like "Consideration is also given to the testing environment, which is sometimes "sterile"...

Lines 236-244: There seem to be different issues interspersed throughout this paragraph; 1) the first is research on effective interventions is limited (though I would argue that plenty of research has been done to examine effective interventions...it just may not have produced effective interventions, or effectiveness for everyone.  2.  That medical professionals lack knowledge of potential options for treatment - that's not necessarily a research/literature issue, it may be an education, dissemination, or resource issue.  The second sentence does not necessarily flow from the first.

Line 256 - PCS previously defined.

Lines 269:  Could you site a few studies that support early neuropsychological intervention following mTBI?

Line 274: In line 245 the authors describe "cognitive, emotional, and physiological symptoms," and here it's "cognitive, emotional, and somatic symptoms."  I'd choose a term for the latter and be consistent

Lines 279-280: "Current research supports..."  can you cite a few articles here?

Line 284:  Do you mean address "psychological research" in particular, or “research” in general as it pertains to persistent PCS?

Lines 288-289: It'd be helpful to add a few citations on the research that is being described.

Line 302 - Please clarify this sentence - firm endorsements of cognitive rehabilitation intervention strategies?

Lines 306-316 provide a nice description of cognitive rehabilitation.  I think it'd be good to include this earlier (e.g., when you introduce it in the preceding paragraph) so that the reader is oriented.

Line 318:  ACRM is previously defined, no need to spell it out here.

Line 363:  Reword "modulate fMRI activity"...perhaps "as well as modulate prefrontal cortex activity[65] as observed through fMRI"

Line 367: Reconsider wording:  instead of "at least screening measures...", how about " screening measures for mental health conditions at a minimum...

Line 397:  Use "PCS" since previously defined

Line 400 - suggest adding a word "....and PROVIDE compensatory strategies..."

Lines 404 - would a better term for "alternative treatment methods" be "complementary and alternative medicine therapies": http://www.mayoclinic.org/healthy-lifestyle/consumer-health/in-depth/alternative-medicine/art-20045267

Line 439: the authors should flesh out a little more what they mean by "heterogeneity across individuals" - in what respect?  Symptom presentation? (which is addressed as the next phrase); demographic characteristics; injury etiology...?

Line 440: "Dearth of quality research examining services for PCS" - this phrase needs to be better defined - what do you mean by this?

Line 445: Instead of "complex and challenging patients," what about "build our understanding of how to best treat patients with complex and challenging conditions" - this separates the patient from his/her symptoms.

Author Response

Manuscript ID: brainsci-210894

Title: Evaluation and Treatment of Mild Traumatic Brain Injury: The Role of Neuropsychology

Brain Sciences Special Issue: Mild Traumatic Brain Injury (mTBI): Medical Treatments and Complications

Authors’ Note:

We thank the reviewers and assistant editor, Annie Zhang, for your time, careful review of our manuscript, and thoughtful and useful feedback, comments, and suggestions.

We have attempted to incorporate your feedback into the current manuscript when possible and have utilized track changes in Microsoft Word to highlight any changes that we have made to the original manuscript. We have not been able to address some of the concerns voiced by both reviewers regarding our overall design. The authors were invited by the guest editors of this Special Issue of Brain Sciences to provide an overview of the role that neuropsychologists can play in the treatment of individuals presenting with a mTBI history. Our intent with this article was not to provide a systematic review or meta-analysis of the current literature. Instead our intent was to provide education regarding the various roles that neuropsychologists can play in the treatment of individuals with mTBI history. With this intent in mind, and in response to some of our feedback, we have made revisions to our original submission that we hope will better convey our intended purpose.  We have also included individualized responses to all of the reviewers’ comments below. Thank you again for your time and consideration of our manuscript.

Responses to Comments:

Reviewer 2 Comments and Responses:

Major conceptual comments: The main concerns are that there should be more discussion about the literature – references should be added where noted, and there needs to be more of an in-depth analysis of what the limitations of the literature are.  The tone of the article seemed quite negative towards research, and may distract from the intended message.  One can critically evaluate the literature by describing the findings and noting its strengths and weaknesses without editorializing.

Response: Thank you for these concerns. As discussed above, the original intent of our manuscript was to provide education regarding the role of neuropsychologists in the treatment of patients presenting with mTBI. We have attempted to add additional references and clarifications where suggested. With regard to the overall tone of the article seeming “quite negative towards research,” the authors are unclear as to what specific instances convey this tone. We have attempted to revise the overall tone of the manuscript as much as possible and hope that the current draft will convey less of a negative tone. Both authors are primarily involved in clinical work in their current roles; however, both have conducted research in the past and continue to engage in research when the opportunity presents itself. We are very aware of the importance of research in informing our clinical work and certainly are not intending to denigrate the work of other researchers in our field. If the manuscript continues to convey this tone in the current revision, we would appreciate additional clarifications of these instances.

Other Comments that must also be addressed and/or considered:

Comment: Abstract, Lines 14-15:  "is briefly reviewed" should be "are" briefly reviewed since the authors are discussing "Complications"

Response: Thanks. Corrected.

Comment: Abstract, Line 17: Delete "unfortunate" - it's implied.

Response: Removed.

Comment: Line 23:  "Estimates across analyses vary, however, it"  - suggested rewording: "Although estimates across analyses vary, it is generally..."

Response: Thanks for the suggestion. Reworded.

Comment: Line 24:  Remove "With that said, these estimates" and instead replace with "These percentages likely underestimate..."

Response: Replaced.

Comment: Line 28:  What's the difference between an mTBI being "unidentified" vs. being "underdiagnosed?"  It would seem like a mild TBI being called a "bump on the head" would be an "underdiagnosis", (but could be unidentified), but not going to the doctor at all is "unidentified" because there's nobody being asked to make an identification.   What I'm inferring from your previous sentence about going to a general practitioner is that they don't have the expertise to diagnose a TBI, and therefore it's "underdiagnosed."  To make your sentences parallel, I would rearrange and say "underdiagnosed or unidentified" or just choose one of those terms depending on your intent.

Response: We agree and have rearranged the sentence to say “underdiagnosed or unidentified.”

Comment: Line 47, Capitalize so that it says "Congress."

Response: Thanks. Corrected.

Comment: Line 74:  Emergency Department can be abbreviated to ED since it's previously defined.

Response: Corrected.

Comment: Line 77: I'd be careful about using the word "failed" because it seems unnecessarily harsh. Instead, I'd say something like "....clinician judgment on mTBI diagnosis was inconsistent with ACRM-based criteria...."

Response: Good point. We have accepted your suggestion.

Comment: Line 91 "Discharged" instead of "Discharge"

Response: Corrected.

Comment: Line 143:  Similar to my previous comment, I would be more neutral in how something is presented.  Instead of saying "suffering" I would say something like "symptoms" or "health condition(s)"

Response: Thanks, the manuscript has been corrected with “symptoms.”

Comment: Line 147:  Similar to previous comments, I'd choose another word to describe this situation.  Instead of saying "failed," I'd say something along the lines of "data not supporting an association" - otherwise it looks like there was something deficient with the researchers.

Response: The manuscript was revised from “failed to show any” to “shown no.”

Comment: Lines 161-169, instead of saying "mTBI patients," I'd say "patients with mTBI history".  That phrase is used several times throughout this paragraph.

Response: Thank you for the reminder on the importance of person-first language. The terminology was changed as suggested throughout the manuscript.

Comment: Lines 170-171 - "Despite the efforts of many researchers attempting to win the controversial debate..." Suggested rewording: "The literature offers different viewpoints on neurogenic versus psychogenic causes of PCS, and to date the reasons for symptom persistence is not fully known."  To say "etiology of symptom persistence" suggests a single source.

Response: Based on this comment and the comments below, we have completed revised this paragraph and hope that it presents a more neutral stance.

Comment: Lines 172-173: I'm not sure if this sentence is needed...it seems like a truism that anyone (even researchers!) would agree that appropriate care is needed for these or any other patients.

Response: The sentence was removed.

Comment: Lines 173-174: The paragraph starts off by talking about bickering researchers who probably agree that patients with PCS should get appropriate care, and then transitions into psychologists saving the day.  I'd suggest that your audience may be turned off by this approach, especially since may psychologists are also researchers.  Eliminate the word "fortunately", and I suggest saying something like "As many psychologists engage in patient-centered care, regardless of symptom etiology, patient-specific factors can be evaluated...."

Response: This paragraph has been completely revised based on feedback.

Comment: Lines 170-176 - I would strongly urge the authors to reconsider their approach to this paragraph.  The beginning of the paragraph makes it seem as if researchers are fighting among themselves trying to prove superiority over an age-old issue that will likely not get resolved anytime soon, and the authors then concede that researchers probably think that people with health conditions should have be treated with the right kind of care.  But, as researchers bicker amongst themselves, fortunately there are psychologists who are above the fray and get to what's important - treating symptoms and taking into account multiple factors so they can make informed decisions about treatment.  The tone of the paragraph makes it seem the authors are pitting researchers against clinicians, and these populations, which at many times complement each other, often also intersect.

Response: Thank you for the feedback. As discussed above, this paragraph has been completely revised. It was by no means our intent to cast researchers in a negative tone. The authors have engaged in research and are very aware of the value that research plays in informing clinical judgment and vice versa. We hope that the revised paragraph presents a less biased and negative tone.

Comment: Line 178:  Add "history" after mTBI

Response: Added.

Comment: Line 224: "Since one impaired test score" sounds a little funny; what about "Since one test score indicative of impairment does not necessarily translate...."

Response: Thanks for the suggestion, it was included in the manuscript.

Comment: Line 226:  Eliminate "to the fact" and state something like "Consideration is also given to the testing environment, which is sometimes "sterile"...

Response: The sentence has been revised with your recommendations.

Comment: Lines 236-244: There seem to be different issues interspersed throughout this paragraph; 1) the first is research on effective interventions is limited (though I would argue that plenty of research has been done to examine effective interventions...it just may not have produced effective interventions, or effectiveness for everyone.  2.  That medical professionals lack knowledge of potential options for treatment - that's not necessarily a research/literature issue, it may be an education, dissemination, or resource issue.  The second sentence does not necessarily flow from the first.

Response:  You are correct that the paragraph did not flow and included two separate issues.  I chose to remove the comment on possible lack of knowledge among medical professionals, as the question of education/dissemination/advocacy is not a central point of this article.  You also make a good point that “limited” was not an appropriate term, and I have rephrased in accordance with the conclusions of the various systematic reviews referenced in this section.

Comment: Line 256 - PCS previously defined.

Response: accepted

Comment: Lines 269:  Could you site a few studies that support early neuropsychological intervention following mTBI?

Response:  This paragraph required reframing; the preceding paragraph summarizes the literature on early intervention in general (with appropriate references), and this paragraph extrapolates from existing research to suggest ways in which psychology/neuropsychology may be especially helpful in the acute phase after mTBI.  This was not clear in the original manuscript and changes have been made to make the incorporation of opinion explicit.  

Comment: Line 274: In line 245 the authors describe "cognitive, emotional, and physiological symptoms," and here it's "cognitive, emotional, and somatic symptoms."  I'd choose a term for the latter and be consistent

Response:  accepted, with “somatic” used consistently

Comment: Lines 279-280: "Current research supports..."  can you cite a few articles here?

Response:  As discussed above, I have related this to the review in the previous paragraph (with references to systematic reviews), and clarified the incorporation of our recommendations.

Comment: Line 284:  Do you mean address "psychological research" in particular, or “research” in general as it pertains to persistent PCS?

Response:  You make a good point that this statement applies to mTBI/PCS research in general. “psychological” has been deleted.

Comment: Lines 288-289: It'd be helpful to add a few citations on the research that is being described.

Response:  References have been added to existing systematic reviews.  The common conclusion across all systematic reviews appears to be that we need more controlled, prospective research with clarity of interventions and consistency of outcomes measured.

Comment: Line 302 - Please clarify this sentence - firm endorsements of cognitive rehabilitation intervention strategies?

Response:  Yes – I have edited the sentence to clarify.

Comment: Lines 306-316 provide a nice description of cognitive rehabilitation.  I think it'd be good to include this earlier (e.g., when you introduce it in the preceding paragraph) so that the reader is oriented.

Response:  Good idea.  I have moved this paragraph to the top of the section.

Comment: Line 318:  ACRM is previously defined, no need to spell it out here.

Response:  accepted

Comment: Line 363:  Reword "modulate fMRI activity"...perhaps "as well as modulate prefrontal cortex activity[65] as observed through fMRI"

Response: accepted

Comment: Line 367: Reconsider wording:  instead of "at least screening measures...", how about " screening measures for mental health conditions at a minimum...

Response:  accepted

Comment: Line 397:  Use "PCS" since previously defined

Response:  accepted

Comment: Line 400 - suggest adding a word "....and PROVIDE compensatory strategies..."

Response:  accepted

Comment: Lines 404 - would a better term for "alternative treatment methods" be "complementary and alternative medicine therapies": http://www.mayoclinic.org/healthy-lifestyle/consumer-health/in-depth/alternative-medicine/art-20045267

Response:  Thank you for the education on appropriate terminology, it has been incorporated.

Comment: Line 439: the authors should flesh out a little more what they mean by "heterogeneity across individuals" - in what respect?  Symptom presentation? (which is addressed as the next phrase); demographic characteristics; injury etiology...?

Response:  Thanks for the clarification.  I had intended this to mean that there is a great deal of variability in development of and recovery from PCS even in the absence of intervention; I have re-ordered and re-worded to clarify.

Comment: Line 440: "Dearth of quality research examining services for PCS" - this phrase needs to be better defined - what do you mean by this?

Response:  Yes, you are right.  I have rephrased as “prospective, controlled studies,” which systematic reviews generally conclude are needed to strengthen the evidence base for PCS intervention.

Comment: Line 445: Instead of "complex and challenging patients," what about "build our understanding of how to best treat patients with complex and challenging conditions" - this separates the patient from his/her symptoms.

Response:  I agree with your feedback on person-first language.  I have reworked that last sentence for simplicity and omitted that phrase.

Round 2

Reviewer 2 Report

Thank you for the opportunity to re-review this paper.  I appreciate the authors outlining the purpose of the manuscript and making edits based on previous comments.    The comments below should be easily addressable.

A.      Conceptual/Major

1.       Lines 162-167:  Symptoms are usually described as subjective and perceived by the patient, whereas signs are more objective and able to be observed by others.  I'd use a  term other than "objective symptoms" (line 162) - perhaps "medical signs" or "objective indicators."  

Likewise, I wouldn't use the term "objective cognitive symptoms" (line 162) since they can't be observed physically.  There may be a better term that the authors can come up with, but I'd say something like "cognitive difficulties that are present shortly after injury" or "cognitive difficulties that appear to result directly from the injury" or "primary cognitive difficulties that are present shortly after injury" or something of that nature.

In lines 164-166, the term "subjective cognitive complaints" is used.  I think the authors are trying to describe "secondary cognitive complaints" to differentiate between primary cognitive symptoms that seem to result from the injury. 

2.       Line 212:  Is cognitive reserve a biological factor?  It's not a "physical" entity, like brain mass or density, but rather a mechanism by which the brain operates, which could be influenced by non-biological factors such as formal education, environmental experiences, learned compensatory strategies, etc.   Do any of these articles specifically list "cognitive reserve" as a biological factor?  If not, I'd be more likely to place it under the psychosocial factors list.  If so, then leave it.

B.      Minor

1.       Lines 97-98: mTBI is already defined, no need to spell it out again.

2.       Lines 111-112: No need to use "CTBIE" since it's only used once.

3.       Line 131:  "do not receive routine follow-up...." - should "evaluation" or "care" be added here?

4.       Lines 169-170:  Just a point of clarification - are the patients reporting more symptoms relative to those without these sensitivities (e.g., 7 vs. 3 symptoms; dichotomous yes/no scale), or are they reporting more severe symptoms (e.g., 7.3 vs. 3.3 on a 1-10 Likert scale)?

5.       Line 209:  "Injury characteristics like mTBI severity" - what is meant by mTBI severity?  Do the authors mean TBI severity (e.g., mild, moderate, or severe), nature of injury (e.g., motor vehicle accident vs. fall vs. other), something else?  Please clarify.

6.       Lines 210-211:  When describing biological factors that are identified in "prolonged mTBI recovery," the authors are implicitly describing a poorer (though not nefgcessarily "poor")  trajectory and should mention direction, since they bring up the term "prolonged."  It's implied that "poorer" genetics would be associated with worse outcomes, but can you speak more about how prolonged recovery is associated with age (older?), gender (female?), prior mTBI history (yes), and physical health (poorer?)?

7.       Line 219:  Does "a more moderate or severe" injury refer to the categorizations of moderate or severe TBI?

8.       Line 238: Could you be more descriptive than "collaterals?":  Family members, teachers, those who know the patient well, others? 

9.       Lines 309-311: Consider rewording:"Consideration is also given to the controlled testing environment, which may not adequately capture the real-world impact of mTBI symptoms....

10.   Line 334:  capitalize "Neurology" and "Rehabilitation"?

11.   Line 338:  Consider "....for months or years FOLLOWING INJURY"

12.   Line 348:  "met ITS standards" rather than "their."

13.   Line 351:  Since "ED" was used before, use that term instead of "emergency room?"

14.   Line 358: Instead of "that account for" what about "that address"?

15.   Line 377: Capitalize Neuropsychology since you're talking about referral to a service?

16.   Line 389:  Instead of "plagued" what about "limited?"

17.   Line 478: add "history" after "mTBI"?

18.   Line 530: mild TBI --> mTBI

19.   Line 577:  mild TBI --> mTBI

Author Response

We would like to thank the reviewer and assistant editor, Annie Zhang, for your time, careful review of our revised manuscript, and thoughtful and useful feedback, comments, and suggestions.  We have incorporated your feedback into the current revision of the manuscript. One point to mention, the lines identified in the reviewer’s comments do not match with the manuscript draft version that we accessed on the website. With this in mind, we have tried to identify the sections of the manuscript that were being referred to; however, we may have made mistakes. Any times when it is unclear, we have made note of it in our individual responses.

Thank you once again for your time and consideration of our revised manuscript.
